# Molecular Detection and Phylogenetic Analysis of Canine Distemper Virus in Marsican Brown Bear (*Ursus arctos marsicanus*)

**DOI:** 10.3390/ani12141826

**Published:** 2022-07-18

**Authors:** Cristina Esmeralda Di Francesco, Camilla Smoglica, Vincenza Di Pirro, Federica Cafini, Leonardo Gentile, Fulvio Marsilio

**Affiliations:** 1Faculty of Veterinary Medicine, University of Teramo, 64100 Teramo, Italy; csmoglica@unite.it (C.S.); federica.cafini@hotmail.com (F.C.); fmarsilio@unite.it (F.M.); 2Veterinary Service, Parco Nazionale d’Abruzzo, Lazio e Molise, Viale Santa Lucia, 67032 Pescasseroli, Italy; vincenza.dipirro@gmail.com (V.D.P.); leonardo.gentile@parcoabruzzo.it (L.G.)

**Keywords:** Marsican brown bear, canine distemper virus, H gene, phylogenetic analysis, wildlife lineage

## Abstract

**Simple Summary:**

Marsican brown bear is a subspecies of Eurasian bear, that lives in a few areas of Central Italy, with an estimated population of only 50 animals. For this reason, it is considered one of the most threatened Italian mammals, and specific Conservation Plans are applied with the focus to fight the mortality causes, mainly related to human activities or illegal practices. On the contrary, few reports describing infectious or parasitic diseases in Marsican brown bears are available. Among pathogens, the canine distemper virus (CDV) is responsible for a contagious and multi-organ disease, able to infect a wide range of domestic and wild carnivores. In 2013 a fatal outbreak of distemper was registered in Central Italy, involving dogs, Apennine wolves, badgers, and foxes, but apparently without any consequences for the Marsican brown bears living in the same territories. In this paper, we describe the first CDV infection detected in a live-trapped bear. The identified strain resulted in similarities to CDV recovered from foxes and dogs of the same area. Even if no clinical signs referred to the disease have been detected in the monitored bear, the evidence of a viral pathogen potentially able to menace the conservation of the Marsican brown bear population highlights the importance of continuing observation activities.

**Abstract:**

In this paper, we report the first molecular detection of the canine distemper virus in the Marsican brown bear (*Ursus arctos marsicanus*). Three subadults and one adult were live-trapped and checked for the main viral pathogens responsible for infectious diseases in this species. The four bears were found to be negative for all investigated viruses except for one, which resulted in a positive outcome for CDV by means of RT-PCR targeting fragments of viral N and H genes. The sequence analysis revealed the specificity of amplicons for the Europe Wildlife lineage of CDV, the same viral strain recovered from three foxes and two unvaccinated dogs coming from the same territories where the positive bear was captured. These results confirm the receptivity of Marsican brown bear for CDV, apparently without any pathological consequences for the positive animal, and suggest the presence in the studied area of a unique wild host-adapted lineage of the virus, able to spread in domestic animals, too. In this respect, continuous and specifically targeted surveillance systems are necessary in order to highlight any changes in the epidemiology of the infection in the territories where the Marsican brown bear lives, along with a more effective vaccination program for domestic dogs co-existing with this endangered species.

## 1. Introduction

Canine distemper virus (CDV) is an enveloped ssRNA member of the genus *Morbillivirus*, family *Paramyxoviridae*, able to cause a highly contagious disease in domestic and wild carnivores [1]. The most common signs of clinically evident distemper include respiratory, gastrointestinal, and neurological symptoms that can evolve in multisystemic fatal forms, particularly in young or non-immunocompetent animals [2]. The host range of CDV is wide, being the virus able to infect different orders and family members, including as well as dogs, wild canids, mustelids, ursids, large felids, and marine mammals, with different onsets of the disease [3].

The genetic characterization of CDV strains is currently based on the different sequences of the hemagglutinin (H) gene, encoding for a structural glycoprotein of the envelope, essential to begin the cellular infection by means of the attachment of the virions to the SLAM (signaling lymphocyte activation molecule) receptor [4]. Until now, 19 different lineages of CDV have been recognized with different temporal, geographical, or host distributions [5]. In Italy, at least three different major European lineages have been documented in both domestic and wild species: Europe-1, originally related to domestic outbreaks but subsequently diversified in wild host-adapted subclades in the Alpine area of Northern Italy [6,7,8]; Europe-2 or Europe-Wildlife viral strains, detected in foxes (*Vulpes vulpes*) [9]; and Europe-3 (namely Arctic lineage), responsible for the distemper outbreak that occurred in 2013 in Central Italy involving feral domestic dogs and wild carnivores [10]. In more detail, the onset of the epidemic started in the Abruzzi region where at least 20 carcasses of Apennine wolves (*Canis lupus*) tested positive for CDV RNA. Six were rescued alive with clinical signs of infection. The area involved in this event is characterized by the presence of the Marsican brown bear (*Ursus arctos marsicanus*), a critically endangered subspecies of the Eurasian brown bear. The population consists of about 50 individuals habiting an extremely small range, essentially limited to the Abruzzo, Lazio, and Molise National Park (ALMNP) area, with frequent incursions into neighboring territories [11]. Indirect exposure of Marsican brown bears to CDV has been documented in the past by means of serological investigations [12,13], although no evidence of active infection and/or clinically relevant manifestation of distemper was reported during the monitoring activities, carried out on the population in accordance with the National conservation plans [14,15].

In 2021, four Marsican brown bears were captured and monitored by the technical staff of the ALMNP in the territories of the Park and surrounding areas, as planned by the Conservation plan [15]. The animals were live-trapped in accordance with international guidelines [16], sexed, and the age was estimated, based on the eruption and consumption of the teeth, considering three broad age classes (cubs, subadults, and adults), as previously reported [17]. All bears were clinically examined by the veterinary staff to highlight any pathological signs or lesions and biological specimens were collected for diagnostic purposes.

Concurrently two adult female red foxes (*Vulpes vulpes*) had been rescued alive in the ALMNP territories, Opi and Barrea municipalities (L’Aquila province, Abruzzi region) showing marked CDV infection-related symptoms, such as neurological signs and oculo-nasal discharge. The animals were kept in isolation for veterinary care in the facilities of the Park and mucosal swabs were collected. Both animals died after the recovery due to the gravity of the symptoms. A third female fox was found dead in the municipality of Pescasseroli (L’Aquila province), near the area of the first recovery.

Finally, a concomitant suspected CDV infection in unvaccinated shepherd dogs has been reported by the owner of a flock of sheep grazing in the same aforementioned territories. Similarly, the ALMNP veterinary staff collected mucosal swabs from two symptomatic dogs for diagnostic investigations.

In this paper, we report the results of diagnostic investigations carried out on the biological samples collected from the aforementioned animals, including the post-mortem examination of the deceased foxes. Direct evidence of CDV infection in foxes, dogs, and for the first time, Marsican brown bear, has been obtained by RT-PCR analysis, and the genetic characterization of viral lineage responsible for the infection was achieved.

## 2. Materials and Methods

The anamnestic data of animals under study, including age, sex and clinical signs, are reported in Table 1, while the recovery sites were showed in the Figure 1. All biological samples were sent to the laboratories of the Veterinary Medicine Faculty of Teramo for further analysis, while two fox carcasses were transferred for necropsy and histopathological investigations to the Istituto Zooprofilattico Sperimentale dell’Abruzzo e del Molise (IZSAM) based in Teramo.

Viral RNA/DNA was obtained from all mucosal swabs by means of MagPurix 12A Nucleic Acid Extraction System (Zinexts Life Science Corp., Taipei, Taiwan), using the MagPurix^®^ Viral/Pathogen Nucleic Acids Extraction Kit B, following the manufacturer’s instructions.

Diagnostic hemi-nested RT-PCR for detection of CDV RNA, able to amplify a 180 bp final fragment of a conservative portion of N gene, was performed as previously reported [18]. Additionally, a biomolecular screening for canine parvovirus (CPV-2), canine adenoviruses (CAdV-1 and CAdV-2), canine herpesvirus (CHV), and canine coronaviruses (CCoVs) was carried out in order to rule out the involvement of other viral pathogens, commonly associated with infectious diseases in carnivores [19,20]. Blood samples recovered from bears were analyzed by virus neutralization test for anti-CDV antibodies titration, as previously described [13].

In accordance with previously published protocols, partial and full-length H gene sequences were amplified from samples that tested positive for diagnostic hemi-nested RT-PCR, [21,22].

Total H gene from a fox and partial fragments of H genes from a dog, along with partial fragments of both N and H genes from a bear, were subsequently purified and submitted for sequencing, using the primers utilized for the amplification.

Nucleotide sequences were analyzed to confirm the specificity for CDV and to compare them with analogous sequences available online, using the CHROMAS software, FASTA (http://www.ebi.ac.uk/fasta33 (accessed on 31 May 2022)), Basic Local Alignment Search Tool (BLAST), and Clustal Omega (http://www.ebi.ac.uk/Tools/msa/clustalo (accessed on 31 May 2022)). A Maximum-likelihood tree of the partial sequences of H gene by means of MEGA software, version 11 [23].

## 3. Results

The results of laboratory investigations were reported in Table 2. All the tested mucosal swabs tested negative for canine viral sequences, except for the CDV N gene fragment, amplified in a total of seven samples.

In detail, five positive samples came from all suspected infected foxes along with a symptomatic dog, while the remaining two samples were obtained from one Marsican brown bear. No amplification was obtained from samples of the remaining bears. All bears tested serologically negative for anti-CDV neutralizing antibodies.

Necropsy examination of foxes revealed a good body condition score of both carcasses with a moderate post-mortem change, mesenteric lymph nodes, and pulmonary parenchyma edema. The histopathological investigation highlighted interstitial pneumonia with foci of purulent bronchopneumonia.

Positive samples were submitted to sequencing of partial (420 bp from dog and bear) and full-length (1824 bp from one fox) CDV H gene, along with a 196 bp portion of N gene obtained from the bear. The sequences were submitted to GenBank with the Access Numbers OM714799-OM714801 for the H gene and OM714802 for the N gene.

The comparison of analogous sequences confirmed the specificity of amplicons for the CDV genome showing a 99–100% of identity with the isolate CDV599/2016 (GenBank Access number KX545421), recovered from a fox found dead in 2016 in the area of L’Aquila province, Abruzzi region. The phylogenetic analysis revealed that the H gene sequences under study clustered in the European wildlife lineage (Europe-2) related to similar sequences from domestic and wild hosts (Figure 2).

## 4. Discussion

In this paper, for the first time, direct evidence of CDV infection in the Marsican brown bear is reported. Canine distemper virus continues to represent a serious threat to the conservation of vulnerable or endangered wild species worldwide. The *Ursidae* family has already been recognized as susceptible to the infection with several reports of serological exposure and/or clinically relevant outbreaks, with particular regard for the captive giant panda (*Ailuropoda melanoleuca*) [24,25,26]. In the *Ursus* genus, the published data were mainly relative to serological investigations while only one report of fatal symptomatic infection was outlined in a wild black bear (*Ursus americanus*) [27,28,29,30,31].

In this respect, the results presented in this study confirm the susceptibility of brown bears to CDV. However, in addition, they allow investigation of the etiological features of the involved viral strain, and the chance to obtain new information about the pathological evolution of the infection in this species.

Noteworthy, the bear with a positive result appeared to be clinically healthy when it was examined by the veterinary staff, suggesting that a poor or null effect has been exerted by the virus on its health status. Probably, this condition can be related to the sampling timing (during incubation or convalescence periods), the low viral load recovered from the animal, or the lineage of CDV responsible for the infection. The virus neutralization failed to highlight any detectable antibody levels, supporting the hypothesis that the infection was at an early stage, rather than an immunosuppressive effect played by the virus. Therefore, an immunosuppressive action of CDV was described in different host species, but this condition appeared to be related to a severe and often fatal onset of the disease. Conversely, a significant increase in VN antibodies was detectable in surviving animals [32]. In this respect, an additional serum sample should be collected from the infected bear in order to highlight if seroconversion has occurred.

The viral strain was characterized as belonging to Europe wildlife lineage, strictly related to viruses detected in symptomatic red foxes and dogs. This relatedness supports the idea that all investigated animals were exposed to a unique viral strain, already described in the same area of study in red fox [33]. Accordingly, the infected bear was captured in Barrea municipality, near the area where the positive red foxes and dogs were recovered.

The Europe wildlife lineage appears to be distinct from the Arctic lineage responsible for the distemper epidemic that occurred in 2013 in the Abruzzo region in both wild and domestic carnivores [10], and it includes other related viral strains recovered from stone marten, mink, badger, and raccoon in Austria, Denmark and Germany [34,35]. In Italy, this lineage is described merely in red foxes, and in this respect, the identification of the same viral strain in the sheep herd dogs could be considered an example of a spill-over from wildlife to a domestic dog. Nevertheless, the European wildlife lineage, originally described in a mink in Denmark [34], has been subsequently reported in domestic dogs, specifically in 2004 in an infected dog in North America [36] and in 2005 in three Hungarian dogs [37], confirming the wide and evolving host range of CDV. The detection of the same viral strain in Marsican brown bear further highlights the interspecies transmission of CDV, particularly favored by the wildlife-domestic interface occurring in peculiar territories of Central Italy, not only in protected areas but in urban and peri-urban environments, too [20,38,39].

The Marsican brown bear is one of the most threatened Italian mammals [40], and despite monitoring and conservation actions carried out over the last decades, the population continues to be stable without evidence of increasing in size. The high levels of human-caused mortality represent one of the main reasons for this trend, along with the small number of reproducing females and a relatively low reproductive rate [11,41]. By contrast, the role of pathogens and related infectious/parasitic diseases seems to be poor or completely absent. Until now, there have only been a few reported cases of infection in the Marsican brown bear; the finding of capillarids eggs, and adults consistent with the genus *Pearsonema* in the bladder of a bear deceased from traumatic gastric rupture [42], a fatal systemic tuberculosis caused by *Mycobacterium bovis* [43] and, a more recent work highlighting the potential role of *Pelodera strongyloides* nematode in the etiology of several cases of dermatitis [17].

Based on the results reported in this study, the evidence of CDV infection in a Marsican brown bear, without any clinically relevant signs of the disease, does not seem to pose a relevant threat to the conservation of this species. However, any changes in the epidemiology of the disease in the territories where the Marsican brown bears coexist with other wild and domestic mammals, able to support the maintenance of CDV in the environment, could lead to a rapid evolution of the infection in the population both in terms of morbidity and mortality rate.

## 5. Conclusions

This is the first direct evidence of CDV infection in the Marsican brown bear species, in absence of distemper-related clinical signs or mortality episodes registered during the study period. Regardless, the small size of the population, along with the numerous threats that can further affect the conservation of this species, make it necessary to put in place specific surveillance programs, focusing on the early identification of pathogens potentially able to influence the health status of these animals. The presence of CDV-infected domestic dogs coexisting with Marsican brown bears once again confirms the necessity to implement the vaccination programs in dog populations, with particular emphasis on those living in protected areas of Central Italy.

## Figures and Tables

**Figure 1 animals-12-01826-f001:**
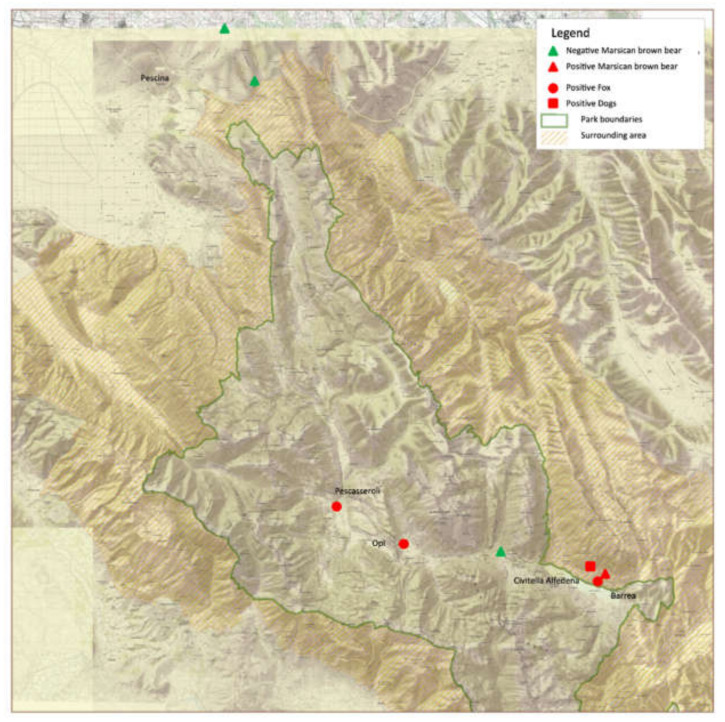
Geographical distribution of animals recovered from ALMNP territories and surrounding localities.

**Figure 2 animals-12-01826-f002:**
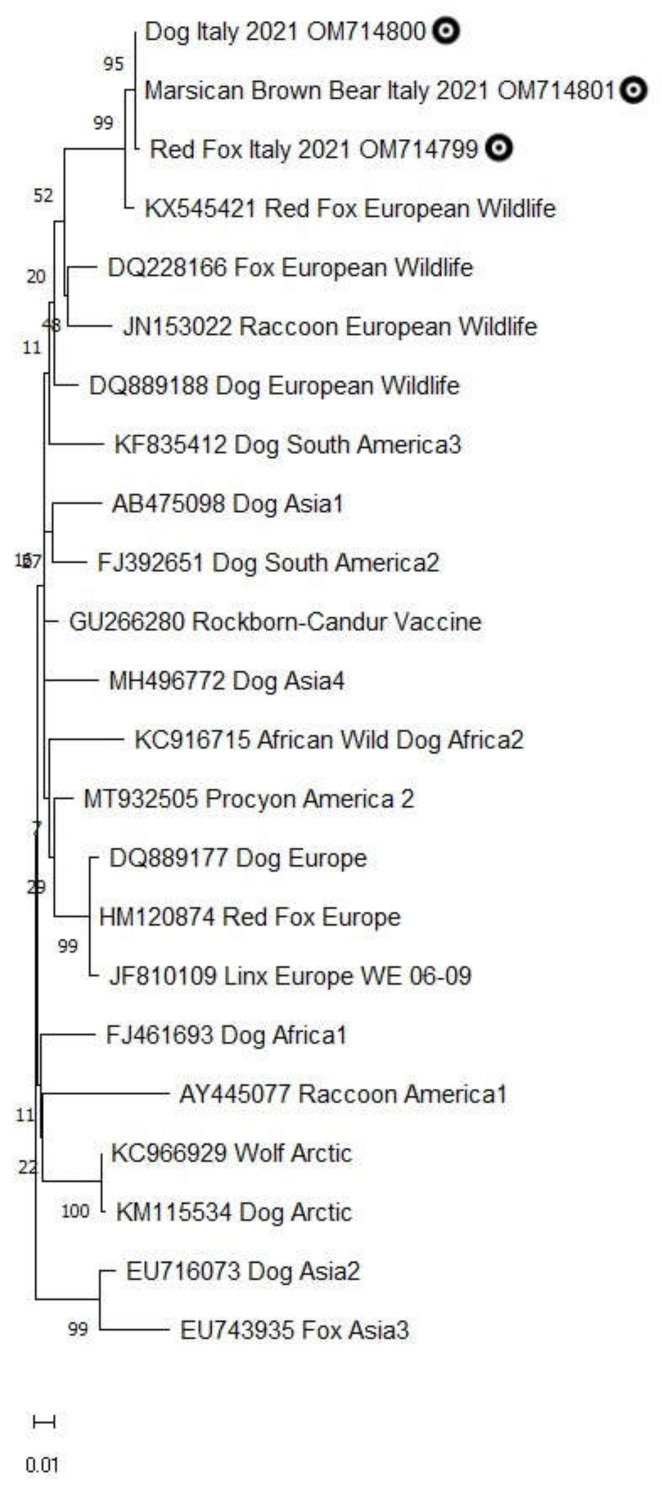
Maximum-likelihood tree obtained analyzing partial sequences of H gene from red fox, Marsican brown bear and dog and analogous sequences representative of main 14 lineages of CDV (Europe, Arctic, European Wildlife, Rockborn Vaccine, America 1 and America 2, Asia 1–4, Africa 1 and Wild Africa 2, South America 2 and South America 3). For each sequence, the GenBank Access number, the host species, and the lineage details are reported. The sequences under study are highlighted with a dark circle. Evolutionary analyses were conducted in MEGA11 [23].

**Table 1 animals-12-01826-t001:** Anamnestic data of investigated animals.

ID Animals	Species	Age	Sex *	Clinical Signs	Area of Recovery (Municipalities)
JC 4621	Marsican brown bear	Subadult	M	Absent	Pescina
GA 4921	Marsican brown bear	Subadult	F	Absent	Pescina
GI 5121	Marsican brown bear	Adult	F	Absent	Civitella Alfedena
RA 5221	Marsican brown bear	Subadult	F	Absent	Barrea
RF 0221	Red Fox	Adult	F	Oculo-nasal discharge, neurological signs	Barrea
RF 0321	Red Fox	Adult	F	Oculo-nasal discharge, neurological signs	Opi
RF 0921	Red Fox	Adult	M	Dead	Pescasseroli
DS 0321	Dog	Adult	M	Neurological signs	Barrea
DS 0421	Dog	Adult	F	Neurological signs	Barrea

* M: male; F: female.

**Table 2 animals-12-01826-t002:** Results of molecular investigations carried out on samples collected from animals under study.

ID Animals	Samples *	CDV	CPV-2	CHV	CAdVs	CCoVs
JC 4621	NS	neg	neg	neg	neg	neg
	RS	neg	neg	neg	neg	neg
GA 4921	NS	neg	neg	neg	neg	neg
	RS	neg	neg	neg	neg	neg
GI 5121	NS	neg	neg	neg	neg	neg
	RS	neg	neg	neg	neg	neg
	VS	neg	neg	neg	neg	neg
RA 5221	NS	**pos**	neg	neg	neg	neg
	RS	neg	neg	neg	neg	neg
	VS	**pos**	neg	neg	neg	neg
RF 0221	NS	**pos**	neg	neg	neg	neg
	RS	**pos**	neg	neg	neg	neg
RF 0321	NS	**pos**	neg	neg	neg	neg
	RS	neg	neg	neg	neg	neg
RF 0921	NS	neg	neg	neg	neg	neg
	RS	**pos**	neg	neg	neg	neg
DS 0321	NS	neg	neg	neg	neg	neg
	RS	neg	neg	neg	neg	neg
DS 0421	NS	**pos**	neg	neg	neg	neg
	RS	neg	neg	neg	neg	neg

* NS: Nasal swab; RS: Rectal swab; VS: Vaginal swab: neg: negative; pos: positive.

## Data Availability

The sequences obtained from this study were submitted to GenBank database with Access Numbers OM714799-OM714802.

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
