# Peer review of "Molecular Detection and Phylogenetic Analysis of Canine Distemper Virus in Marsican Brown Bear (Ursus arctos marsicanus)"

_animals, 2022, doi:10.3390/ani12141826_

Round 1

Reviewer 1 Report

In this manuscript, the authors describe CDV infection in Marsican brown bear, a critically endangered wild species for the first time and the details in molecular characteristics based on phylogenetic analysis. It suggests the interspecies transmission of CDV occurred in the wildlife-domestic interface in peculiar territories of Italy. Despite this study contains a limited study data, it allows to update the host range of CDV and molecular characteristics. Nevertheless, I have some minor issues should be addressed before publication.

1.     The Neighbor-Joining method was adopted to construct the phylogenetic tree, but the Bootstrap value is not high enough. And the virus lineages of sequences retrieved from the GenBank and isolated this study were not presented obviously. Moreover, the maximum-likelihood method maybe a better method for this study.

2.     The results section should be written in paragraphs according to the relevant content, not just one paragraph.

3.     Line 145, “RNA amplification” is inaccurate, since the RNA could not be used as template directly to amplify gene fragments.

4.     Line 212, a full stop was missed at the end of the sentence.

Author Response

Reviewer 1

Comments and Suggestions for Authors

In this manuscript, the authors describe CDV infection in Marsican brown bear, a critically endangered wild species for the first time and the details in molecular characteristics based on phylogenetic analysis. It suggests the interspecies transmission of CDV occurred in the wildlife-domestic interface in peculiar territories of Italy. Despite this study contains a limited study data, it allows to update the host range of CDV and molecular characteristics. Nevertheless, I have some minor issues should be addressed before publication.

  1. The Neighbor-Joining method was adopted to construct the phylogenetic tree, but the Bootstrap value is not high enough. And the virus lineages of sequences retrieved from the GenBank and isolated this study were not presented obviously. Moreover, the maximum-likelihood method maybe a better method for this study.

As suggested by the Reviewer, we replaced the phylogenetic tree with a new tree obtained applying the maximum-likelihood method and 1000 replicates as Bootstrap value. The sequences submitted to Genbank are not yet available online, but if necessary we can send to the Reviewer all sequences obtained in this study.

  1. The results section should be written in paragraphs according to the relevant content, not just one paragraph.

The Results section has been appropriately divided into paragraphs.

  1. Line 145, “RNA amplification” is inaccurate, since the RNA could not be used as template directly to amplify gene fragments.

As correctly remarked by the Reviewer we deleted the term RNA

  1. Line 212, a full stop was missed at the end of the sentence.

Done

Reviewer 2 Report

The manuscript describes the first report of canine distemper virusin a Marsican brown bear. This is of interest to those in the research field. However,

 The paper is very badly written with very poor English and needs extensive rewriting with input from a natural English speaker. There are other points which need to be addressed:

1. Most of the first 3 paragraphs of he materials and methods are introduction and should be moved to the introduction.

2. Line 191. The authors need to explain better. Do they mean that infection must be in the first 2 weeks before the animal seroconverted? Was there any opportunity to take a later blood sample to see if seroconversion had occured? Lack of antibodies might also be due to immunosuppression but less likely as there would probably be some response.  

Author Response

Reviewer 2

Comments and Suggestions for Authors

The manuscript describes the first report of canine distemper virus in a Marsican brown bear. This is of interest to those in the research field. However,

 The paper is very badly written with very poor English and needs extensive rewriting with input from a natural English speaker. There are other points which need to be addressed:

The manuscript has been revised by a native English speaker, as suggested by the Reviewer.

  1. Most of the first 3 paragraphs of he materials and methods are introduction and should be moved to the introduction.

As suggested by the Reviewer, the first anamnestic description of animals included in this study was moved to the introduction and the aim of the paper was modified consequently.

  1. Line 191. The authors need to explain better. Do they mean that infection must be in the first 2 weeks before the animal seroconverted? Was there any opportunity to take a later blood sample to see if seroconversion had occured? Lack of antibodies might also be due to immunosuppression but less likely as there would probably be some response.  

We appreciated the highly valuable comments of the Reviewer regarding the serological findings observed in infected bear. In accordance with previous studies, the immunosuppressive effect of CDV (characterized by lymphopenia, suppression of lymphocyte proliferation and absence of antibodies response) is generally associated to a severe and often fatal clinical onset of the infection (see Zhao J, Shi N, Sun Y, Martella V, Nikolin V, Zhu C, Zhang H, Hu B, Bai X, Yan X. Pathogenesis of canine distemper virus in experimentally infected raccoon dogs, foxes, and minks. Antiviral Res. 2015 Oct;122:1-11). The apparently healthy status exhibited by the infected bear, along with the seronegative result obtained by the VN test can be considered consistent with the hypothesis that the infection was at an early stage. As correctly suggested by the Reviewer, this hypothesis could be confirmed by a further blood sampling of this animal. Unfortunately, until now no additional capture activities have been made and the seroconversion of the bear cannot be confirmed. Anyway, in the discussion section the paragraph relative to this point was improved and the appropriate reference was added (lines 229-240).

Round 2

Reviewer 2 Report

The authors have made revisions to the manuscript as requested, including English editing.